# Assessment of Two-Vehicle and Multi-Vehicle Freeway Rear-End Crashes in China: Accommodating Spatiotemporal Shifts

**DOI:** 10.3390/ijerph191610282

**Published:** 2022-08-18

**Authors:** Chenzhu Wang, Yangyang Xia, Fei Chen, Jianchuan Cheng, Zeng’an Wang

**Affiliations:** 1School of Transportation, Southeast University, 2 Sipailou, Nanjing 210096, China; 2School of Transportation, Tibet University, Lhasa 850001, China; 3Jiangsu Expressway Company Limited, Nanjing 210049, China

**Keywords:** injury severity, spatiotemporal stability, freeway rear-end crashes, random parameters logit model

## Abstract

Accounting for the growing numbers of injuries, fatalities, and property damage, rear-end crashes are an urgent and serious topic nowadays. The vehicle number involved in one crash significantly affected the injury severity outcomes of rear-end crashes. To examine the transferability and heterogeneity across crash types (two-vehicle versus multi-vehicle) and spatiotemporal stability of determinants affecting the injury severity of freeway rear-end crashes, this study modeled the data of crashes on the Beijing-Shanghai Freeway and Changchun-Shenzhen Freeway across 2014–2019. Accommodating the heterogeneity in the means and variances, the random parameters logit model was proposed to estimate three potential crash injury severity outcomes (no injury, minor injury, and severe injury) and identify the determinants in terms of the driver, vehicle, roadway, environment, temporal, spatial, traffic, and crash characteristics. The likelihood ratio tests revealed that the effects of factors differed significantly depending on crash type, time, and freeway. Significant variations were observed in the marginal effects of determinants between two-vehicle and multi-vehicle freeway rear-end crashes. Then, spatiotemporal instability was reported in several determinants, including trucks early morning. In addition, the heterogeneity in means and variances of the random parameters revealing the interactions of random parameters and other insignificant variables suggested the higher risk of determinants including speeding indicators, early morning, evening time, and rainy weather conditions. The current finding accounting for spatiotemporal instability could help freeway designers, decision-makers, management strategies to understand the contributing mechanisms of the factors to develop effective management strategies and measurements.

## 1. Introduction

Currently, traffic crash leads to a great number of incapacitating injuries and fatalities, along with property damages, posing tremendous economic and emotional burden on society. Roadway traffic caused 1.35 million deaths in 2016, and the death rate in low-income countries remained 3 times higher than that in high-income countries [1]. As one of the most prevalent traffic crash types, rear-end crashes result in a large number of severe injuries and fatalities. The main causes of rear-end crashes are attributed to careless driving and close car-following behaviors [2], where the rear side of the front vehicle is hit by the front side of the following vehicle [3]. In the U.S., almost 3.41 million rear-end crashes occurred in 2020, involving 3000 fatalities and 1.386 million injuries [4]. In China, traffic crashes caused 63,772 fatalities in 2017, of which 9.6% suffered from rear-end crashes, along with 16,409 injuries [5]. Therefore, special research efforts should be devoted to exploring the internal mechanisms of and identifying contributing factors to freeway rear-end crashes, and to put forward effective measurements for minimizing the risk and severity of rear-end crashes.

Recent research efforts confirmed that a variety of indicator variables in terms of driver, vehicle, roadway, environmental, temporal, etc., characteristics have been used to analyze the injury-severity outcomes of rear-end crashes (a specific literature review on findings regarding rear-end crashes will be discussed later in Section 2.1) [6,7]. For instance, Zhang and Hassan [8] concluded that driving at nighttime during weekends is highly related to injury and fatal rear-end crashes. Furthermore, rainfall conditions tended to increase the injury-severity levels of rear-end crashes [9]. In addition, the number of vehicles involved in a crash has been significantly identified to be a major precursor in increasing the risk propensity and the resultant injury severity levels of rear-end crashes [10,11], while the rear-end crashes classified by this element have not been examined at a disaggregate level. Despite these efforts, it cannot be denied that explicit understanding of unobserved heterogeneity and spatiotemporal variations in rear-end crashes, particularly those classified as two-vehicle and multi-vehicle crashes, is insufficient.

Unobserved heterogeneity has been examined by a growing body of studies, which might result from the unavailability of factors reported in the police-reported data [12]. Regarded as an essential factor, unobserved heterogeneity leads to variations in the crash injury level of individual drivers due to different physical and mental abilities, perceptions of risk, and reactions to potential hazards [6,13,14,15]. To account for the unobserved heterogeneity across groups and observations, the random parameters approach and its extension have been proposed in recent research efforts [12,16,17]. Whereas the random parameter logit models with heterogeneity in means and variances indicated their superiority and accuracy by accounting for unobserved heterogeneity, leading to biased estimation results [7,18], which will be proposed in this study.

Given the research efforts mentioned above, the overall intent of this study is to explore the spatiotemporal instability in the contributing factors to the injury severity levels of freeway rear-end crashes, along with the different internal mechanisms of two-vehicle and multi-vehicle rear-end crashes. Then, it is critical to gain a better understanding of by what degrees the effects of contributing factors of rear-end crashes will change over time and space, which could be useful for roadway designers, decision makers, and freeway management to facilitate proper countermeasures. Hence, the authors are particularly interested in the following problems: (i) How did the contributing factors to the injury-severity outcomes vary in the two-vehicle and multi-vehicle rear-end crashes? How did the effects of these variables change across the crash type? (ii) Are the effects of determinants stable, temporally or spatially, in two-vehicle and multi-vehicle rear-end crashes? Whether the models can be transferable to other freeways in China? (iii) What contributions do spatiotemporal instability and related findings make? How can countermeasures address the spatiotemporal instability, targeted at different time periods and freeways?

Thus, random parameters logit models with heterogeneity in means and variances were proposed to examine the contributing factors using crash data from Beijing-Shanghai Freeway and Changchun-Shenzhen Freeway across 2014 to 2019. The remainder of this study is structured as follows. Section 2 summarizes the previous research efforts analyzing the rear-end crashes and reviews the approaches accounting for unobserved heterogeneity. Section 3 presents the descriptions of crash dataset. The proposed methodological method is shown in Section 4. Section 5 demonstrates the transferability test results. Section 6 illustrates the discussions and interpretations of the estimated results, followed by conclusions shown in Section 7.

## 2. Literature Review

### 2.1. Literature Review on Previous Research Efforts Analyzing Rear-End Crashes

A large body of research effort has been proposed to examine the contributing factors to injury severity outcomes in rear-end crashes. Table 1 l summarizes the findings in previous research efforts regarding rear-end crashes, while the contributing factors are classified into six categories: driver, vehicle, roadway, environmental, temporal, and traffic characteristics. Several inconsistent findings were reported in previous research efforts. Take the number of vehicles. For instance, two-vehicle crashes were reported by Chen et al. [19] to be the most common rear-end type leading to fatalities. In contrast, Yu et al. [7] illustrated that the two-vehicle indicator was related to a lower possibility of severe injuries. Most previous research efforts focused on rear-end crashes that occurred in work zones or on rural roadways, with only a few studies focusing on freeway rear-end crashes. Furthermore, the recent study tends to examine the injury severities of rear-end crashes that occurred in China’s freeways to explore and elucidate the potential differences in crash mechanisms driven by different driving environments.

### 2.2. Literature Review on Approaches for Unobserved Heterogeneity

Table 2 summarizes the methodological approaches proposed to examine the injury severity of rear-end crashes. These approaches can be classified into two major classifications: statistical methods and data-driven methods; and possible trade-offs always exist between predictive and inference capabilities [25]. Overall, the statistical methods indicated the superiority of explaining potential associations between the contributors and resultant injury severities, while the data-driven methods are more capable of predicting the injury outcomes in the crashes. Previous research efforts have reviewed comprehensive studies accounting for unobserved heterogeneity and temporal stability. (As shown in the review works of Mannering [26], the temporal instability refers to whether the effects of explanatory variables remain stable over time in psychology, neuroscience, economics, and cognitive science. Furthermore, the issues have been confirmed by multiple accident-data analyses, which are not associated with Lapunov stability.) [12,26]. As an extension of the random-parameters methods, the approaches accounting for heterogeneity in the means and variances can provide much more flexibility in tracking the unobserved heterogeneity, showing statistical superiority in terms of accuracy and reduced heterogeneity [13,16,18]. Furthermore, temporal instability has been confirmed by an abundance of research efforts as significant crash contributors can be inconsistent in different years. Ignoring such instability might lead to inadequate estimated results, erroneous conclusions, and ineffective or even dangerous safety policies [26]. Behnood and Mannering [20] demonstrated the existence of temporal instability in the effects of factors influencing injury severity in large-truck vehicle crashes across time of day and year. Islam and Mannering [27] demonstrated significant differences in driver-injuries between aggressive and non-aggressive driving, while the effects of the determinants changed significantly over time. Nevertheless, other than temporal stability, spatial variation remains another problem that cannot be underestimated in safety research. Recent research efforts have proved that biased estimation results might also be attributed to regional differences producing potential heterogeneity in the spatial transferability issues in crash analysis [18,26,28,29], incorporating freeway segments [30], urban arterials [31], urbanized areas [32], counties [33], provinces [34], etc. It is worthwhile to pay additional attention to the effects of contributing factors on injury-severities through integrating temporal and spatial dimensions simultaneously.

## 3. Data Description

Data available in this study were collected from two four-lane national freeways over 2014–2019:The Beijing-Shanghai Freeway (G2) in Jiangsu Province is 259.5 km long and has a design speed of 120 km/h; (ii) the Changchun-Shenzhen Freeway (G25) in Guangzhou Province is 232.7 km long and has a design speed of 100 km/h. Table 3 and Figure 1 reported three injury severity classifications in the dataset: severe injury, minor injury, and no injury (property damage only). It should be noted that this study intends to explore the temporal variations in freeway rear-end crashes, and the dataset was split into three sub-datasets: 2014–2015, 2016–2017, and 2018–2019. Table A1 and Table A2 presented the variables for rear-end crashes on the G2 and G25 Freeways, respectively.

## 4. Methodology

This paper utilized the random parameters multinomial logit model with heterogeneity in the means and variance to estimate the potential heterogeneity and spatiotemporal stability in the contributing factors affecting the injury severity of two-vehicle and multi-vehicle freeway rear-end crashes. Considering three injury severity levels, including severe injury, minor injury, and no injury, the utility function determining the probability density is specified as [14,40]:(1)$$S_{ij}=\beta_{i}X_{ij}+\varepsilon_{ij}$$
where $S_{ij}$ represents a function determining the probability of injury severity outcome i in a rear-end crash j, $X_{ij}$ is a vector of explanatory variables, $\beta_{i}$ is the estimated parameter, while $\varepsilon_{ij}$ is an assumption of the error term following the generalized extreme value distribution. The adopted level of significance in the current study is 95%.

Supposing that $\varepsilon_{ij}$ follows the extreme-value distribution, a standard multinomial logit model can be defined as:(2)$$P_{ij}=\int^{\;}\frac{e^{\beta_{i}X_{ij}}}{\sum^{\;}e^{\beta_{i}X_{ij}}}f(\beta |\varphi )d\beta$$
where f(β|φ) denotes the probability density function of the random vector β, and φ represents the mean and variance, determining the parameters of the probability density function.

Based on Seraneeprakarn et al. [41], the random parameters accounting for heterogeneity in the mean and variance are specified as:(3)$$\beta_{ij}=\beta_{i}+\delta_{ij}M_{ij}+\sigma_{ij}e^{\omega_{ij}D_{ij}}\nu_{ij}$$
where $M_{ij}$ and $D_{ij}$ represent vectors capturing heterogeneity in means and standard deviation $\sigma_{ij}$ with corresponding parameter vector $\omega_{ij}$ for injury severity i in crash j, respectively, $\delta_{ij}$ is a corresponding vector of the estimable parameters, while $\nu_{ij}$ is a disturbance term. The $M_{ij}$ and $D_{ij}$ characterize the attributes of heterogeneity regarding the driver, vehicle, roadway, environmental, temporal, spatial, traffic, and crash characteristics. If the random-parameters logit model is significant in the vector of $M_{ij}$ and $D_{ij}$, the model characterizes the unobserved heterogeneity in the means and variances. If no variables are statistically significant in the vector $M_{ij}$, the model represents only heterogeneity in means.

During model estimation, various density functions (e.g., uniform, lognormal, and triangular) have been evaluated for the distributions of the random parameters. Then the normal distribution can provide a better statistical fit than others (consistent with past research efforts such as Milton et al. [17], Behnood and Al-Bdairi [41], which was utilized in this current study). The Halton sampling approach was proposed in this paper to optimize prediction performance and efficiency [42]. After utilizing a simulated maximum likelihood approach with different Halton draws, the 1000 Halton draws were proposed in the parameter estimation after the trade-off between computation simulation efficiency and estimation performance [41].

The log-likelihood function (LL) is more convenient than original likelihood functions to work with when taking the derivative of a function and solving for the parameter being maximized, which was utilized in this study [17] to examine the model estimation. It is defined as:(4)$$LL=\sum_{n=1}^{N}\left(\sum_{i=1}^{I}\sigma_{ij}\left[\beta_{i}X_{ij}-LN\sum_{\forall I}e^{\beta_{I}X_{Ij}}\right]\right)$$
where I denotes the total number of injury severity outcomes, and other notations and symbols are as defined previously.

## 5. Transferability Tests

A large body of research shows that the effects of contributing factors can change over time [26,27] and vary in different spaces [18,28,29], demonstrating temporal and spatial instability. From a spatial perspective, transferability is desirable because it means that parameters of models estimated in other places can be used, thus saving the cost of additional data collection and estimation. Temporal transferability ensures that forecasts made with the model have some validity in that the estimated parameters are stable over time [26].

This study proposed five series of likelihood ratio tests (LRT), including two series of tests for temporal stability and two for spatial stability, followed by another series of transferability tests across two-vehicle and multi-vehicle freeway rear-end crashes.

The first temporal stability tests were utilized to explore whether the parameter estimation for two individual years remained the same across these years [14]:(5)$$\chi_{t_{1}}^{2}=-2\left[LL\left(\beta_{y_{1}y_{2}}\right)-LL\left(\beta_{y_{1}}\right)\right]$$
where, $LL\left(\beta_{y_{1}y_{2}}\right)$ represents the log-likelihood at convergence of the model incorporating parameters from $y_{2}$ while using data from subgroup $y_{1}$ (2014–2015, 2016–2017 and 2018–2019), while $LL\left(\beta_{y_{1}}\right)$ denotes the log-likelihood at convergence of the model using subgroup $y_{1}$’s data.

The $y_{1}$ subgroup and $y_{2}$ subgroup can be reversed to provide two test results for comparison. The resulting $\chi^{2}$ value under the $X^{2}$ distribution (with degrees of freedom equal to the number of estimated parameters in model $\beta_{y_{1}y_{2}}$ [14]) can be used to explore whether the null hypothesis that the parameters are equal between two year-period data can be accepted or rejected at the confidence level [13]. The likelihood ratio test results of the null hypothesis in G2 models for two-vehicle and multi-vehicle freeway rear-end crashes, respectively, have been presented in Table 4 and Table 5. While Table 6 and Table 7 showed the results for G25 models. Specifically, in G2 two-vehicle rear-end crash models, using the converged parameters of the 2016–2017 model as the starting values and applying them to 2014–2015 data gave an $\chi^{2}$ value of 82.20 with 15 degrees of freedom, showing that the null hypothesis under 99.99% confidence level that the two time periods remain the same can be rejected. Overall, these results demonstrated that the null hypothesis that under >99% confidence the different year tested produced equal parameters can be rejected, which is consistent with recent literature works analyzing the injury severity by different time periods [13,14,20].

Moreover, the temporal stability was examined by utilizing another series of likelihood ratio tests between the joint and each individual model [13,27]:(6)$$\chi_{t_{2}}^{2}=-2\left[LL\left(\beta_{2014-2019,g,s}\right)-\sum_{2014}^{2019}LL(\beta_{t,g,s}\right)]$$
where $LL\left(\beta_{2014-2019,g,s}\right)$ represents the convergent log-likelihood in the model for crash group g (two-vehicle, multi-vehicle freeway rear-end crashes) on the freeway s (G2, G25) for the three period combinations, while $LL\left(\beta_{t,g,s}\right)$ denotes the convergent log-likelihood in the models for crash group g in the freeway s adopting only one period t (2014–2015, 2016–2017, and 2018–2019) data. For two-vehicle and multi-vehicle freeway rear-end crashes, the G2 model estimates give an $\chi^{2}$ values of 268.95 and 215.68 under 38 and 36 degrees of freedom (The degrees of freedom equal to the summation of statistically significant parameters in each year minus the number of statistically significant parameters in the overall model, which are the same in Equations (6)–(8).). The G25 model estimates gave $\chi^{2}$ values of 184.68 and 226.74 with 32 and 31 degrees of freedom, respectively, for two-vehicle and multi-vehicle freeway rear-end crashes. These results also indicated that the null hypothesis temporal stability of separate G2 and G25 models can be rejected under 99.99% confidence.

The following two series of likelihood ratio tests were developed to estimate the spatial stability across the G2 and G25 models. The first one can be implemented with:(7)$$\chi_{s_{1}}^{2}=-2\left[LL\left(\beta_{joint,g,t}\right)-LL\left(\beta_{G2,g,t}\right)-LL\left(\beta_{G25,g,t}\right)\right]$$
where, $LL\left(\beta_{joint,g,t}\right)$ represents the convergent log-likelihood of the model for crash group g (two-vehicle, multi-vehicle freeway rear-end crashes) on the two freeways in year t (2014–2015, 2016–2017, and 2018–2019), $LL\left(\beta_{G2,g,t}\right)$, and $LL\left(\beta_{G25,g,t}\right)$ denotes the convergent log-likelihood in the models for the crash group g in the two freeways in year t, respectively. Then, this study obtained six test results, which could be adopted to examine the transferability of estimated G2 and G25 crash models. For two-vehicle (multi-vehicle) freeway rear-end crashes, the $\chi^{2}$ gave test results of 187.95 (148.51), 119.62 (165.84), and 106.87 (194.85) with 32 (25), 24 (31), and 23 (33) degrees of freedom in 2014–2015, 2016–2017, and 2018–2019 respectively, specifying that the null hypothesis that G2 and G25 models are the same can be rejected with >99.99% confidence.

Otherwise, the second series of spatial transferability can be specified as:(8)$$\chi_{s_{2}}^{2}=-2\left[LL\left(\beta_{G2_{g,t},G25_{g,t}}\right)-LL\left(\beta_{G2_{g,t}}\right)\right]$$
where $LL\left(\beta_{G2_{g,t},G25_{g,t}}\right)$ represents the convergent log-likelihood of the model containing parameters for crash group g (two-vehicle, multi-vehicle freeway rear-end crash) from G25 dataset in year t (2014–2015, 2016–2017, and 2018–2019), and $LL\left(\beta_{G2_{g,t}}\right)$ denotes the convergent log-likelihood of the model adopting the G2’s data in year t. Then, the test was also conducted by reversing the G2 subgroup and the G25 subgroup to obtain two test results for each model comparison. Using G2’s data to fit the parameters of G25’s model, the $\chi^{2}$ test gave results of 168.97 (115.68), 129.68 (154.25), and 129.84 (135.14) with 15 (8), 9 (13), and 9 (10) degrees of freedom for two-vehicle (multi-vehicle) freeway rear-end crashes in 2014–2015, 2016–2017, and 2018–2019, respectively. Otherwise, the $\chi^{2}$ test results using G25’s data in G2 model gave values of 2014–2015, 2016–2017, and 2018–2019 were 187.35 (129.65), 162.38 (135.68), and 116.84 (108.94) with 18 (10), 14 (12), and 9 (8) degrees of freedom for two-vehicle (multi-vehicle) freeway rear-end crashes, respectively. These results both indicate that the null hypothesis for both the G2 and G25 models are the same and can be rejected with >99.99% confidence. This result is also consistent with previous studies analyzing the injury severity model across different spaces, including rural highways [43] and states [29].

Lastly, transferability tests across two-vehicle and multi-vehicle freeway rear-end crashes can be estimated as [27,41]:(9)$$\chi_{g}^{2}=-2\left[LL\left(\beta_{joint,t,s}\right)-LL\left(\beta_{two-vehicle,t,s}\right)-LL\left(\beta_{multi-vehicle,t,s}\right)\right]$$
where $LL\left(\beta_{joint,t,s}\right)$ represents the convergent log-likelihood of the model with both the two-vehicle and multi-vehicle freeway rear-end crashes of the freeway s in year t (2014–2015, 2016–2017, and 2018–2019), and $LL\left(\beta_{two-vehicle,t,s}\right)$ ($LL\left(\beta_{multi-vehicle,t,s}\right)$) represents the convergent log-likelihood of two-vehicle (multi-vehicle) freeway rear-end crashes model on the freeway s in year t. According to this, 12 test results were obtained to test the transferability of estimated models across two-vehicle and multi-vehicle freeway rear-end crashes models. The $\chi^{2}$ test results for two-vehicle (multi-vehicle) freeway rear-end crashes in G2 model of 2014–2015, 2016–2017, and 2018–2019 were 198.51 (158.68), 169.84 (128.48), and 126.87 (157.98) with 18 (14), 15 (12), and 12 (14) degrees of freedom, respectively. Regarding the G25 model, the $\chi^{2}$ results for two-vehicle (multi-vehicle) freeway rear-end crashes were 185.68 (154.51), 152.85 (163.41), and 165.47 (147.85) with 17 (14), 14 (15), and 15 (14) degrees of freedom, respectively. The results demonstrated that the null hypothesis that the truck-involved and non-truck-involved crashes model remain the same should be rejected under >99.99% confidence.

Above all, the transferability test results indicated the temporal instability, spatial instability, and transferability test across different crash types.

## 6. Results and Discussion

Table A3 indicates the estimated result comparisons among four models: the base multinomial logit model (MNL), the random parameters logit model (RPL), the random parameters logit model with heterogeneity in the mean (RPLM), and the random parameters logit model with heterogeneity in the mean and variances (RPLMV). The good-of-fitness is compared based on the $\rho^{2}$ values and the $\chi^{2}$ test [44,45]. The $\rho^{2}$ values and $\chi^{2}$ test results can reflect that the random parameter logit model with heterogeneity in the means and variances (RPLMV) outperformed the other three models (with higher $\rho^{2}$ values and over 95% confidence interval to reject the null hypothesis that the RPLMV remains the same as the other three models), while the subsequent discussion will be utilized based on the RPLMV.

Based on the RPLMV, the estimation results for two-vehicle and multi-vehicle rear-end crashes in the years 2014–2019 were displayed respectively in Table A4 and Table A5. The two-vehicle freeway rear-end model obtained $\rho^{2}$ values of 0.703, 0.655, and 0.640 for years 2014–2015, 2016–2017, and 2018–2019, respectively. The multi-vehicle freeway rear-end model reported $\rho^{2}$ values of 0.709, 0.657, and 0.711 in the corresponding years. The results demonstrated that most of the variables stayed spatiotemporal instable and non-transferable between two-vehicle and multi-vehicle freeway rear-end crashes. To distinctly examine the spatiotemporal instability, the marginal effects of variables pertaining to the injury outcomes were summarized in Table A6 and Table A7.

### 6.1. Driver Characteristics

Showing spatiotemporal instability, safety (1 if speeding, 0 otherwise) was only statistically significant in the 2016–2017 G2 two-vehicle rear-end model (estimated parameter of −0.529 in Table A4). Consistent with previous research [8], the speeding tended to increase the injury levels (decreased likelihood of no injury and increased probabilities of minor and severe injury).

Moreover, safety (1 if improper action, 0 otherwise) was not significant in any of the models.

### 6.2. Vehicle Characteristics

For vehicle types, passenger cars, trucks, and heavy trucks were identified as significant variables determining injury severity levels.

As shown in Table A5 and Table A7, the passenger car was only significant in 2016–2017/2018–2019 G25 multi-vehicle freeway rear-end crashes, showing consistent effects on no injury and minor injury levels (positive marginal values on no injury and negative marginal values). However, the passenger car indicator increased the severe injury likelihood by 1.26% in 2016–2017 while decreasing the corresponding likelihood by 0.15% in 2018–2019 (see Table A7).

Inconsistent values also existed in the effects of trucks on minor injuries, whereas this variable tended to increase the severe injury likelihood in 2014–2015/2018–2019 G2 multi-vehicle (marginal effects of 0.0054/0.0665 in Table A6) and 2014–2015 G25 two-vehicle/multi-vehicle freeway rear-end crashes (marginal effects of 0.0253/0.0057 in Table A7). This finding might be attributed to the greater crash aggressiveness of the trucks [46,47], which will give rise to greater hazards for other vehicles.

Heavy trucks tended to increase the severe injury likelihood in G25 2014–2015 two-vehicle and 2016–2017/2018–2019 multi-vehicle freeway rear-end crashes (see Table A7). Nevertheless, this vehicle type increased the minor injury likelihood by 4.91%, while it increased the no injury and severe injury likelihood by 4.68% and 0.23%, respectively, in G2 2018–2019 two-vehicle freeway rear-end crashes (see Table A5).

### 6.3. Roadway Characteristics

With regard to roadway characteristics, $R_{front}$, $L_{front}$, $R_{present}$, $L_{present}$, $Ls_{max}$, and $Ls_{min}$ were identified as determinants significantly affecting injury crash severity.

The $R_{front}$ was shown in Table A7 to reduce the likelihood of severe injury in 2018–2019 G25 multi-vehicle freeway rear-end crashes. Likewise, $R_{present}$ tended to decrease the possibility of severe injury by 2.88% and 2.57%, respectively, in G2 2016–2017 two-vehicle and multi-vehicle freeway rear-end crashes (see Table A5). The possible explanation might be that the greater radius was related to greater stopping sight distance (SSD) [48], which enables drivers to perceive potential hazardous materials and operate more properly [49].

Suggesting potential spatiotemporal instability and heterogeneity by vehicle number, $L_{front}$ is also found in Table A5 to be significant in 2017–2017 G25 multi-vehicle freeway rear-end crashes. Other than G2 2016–2017 multi-vehicle freeway rear-end crashes, $L_{present}$ yielded an increased possibility of severe injuries in 2014–2015/2016–2017 G2 two-vehicle (marginal effects of 0.0766/0.0149 in Table A6) and 2018–2019 G25 two-vehicle/multi-vehicle freeway rear-end crashes (marginal effects of 0.0006/0.0368 in Table A7). A possible explanation is that drivers with less operation and driving workload are prone to drowsiness and hypervigilance under monotonous driving patterns when approaching the same curvature [50,51].

The $Ls_{max}$ was negatively related to minor/severe injury likelihood of 2016–2017 G2 two-vehicle freeway rear-end crashes (marginal effects of 0.0406/0.0149 in Table A6), specifying potential spatiotemporal instability and heterogeneity by crash type. The $Ls_{min}$ was linked to a decrease in no injury-likelihood and increased minor-injury likelihood.

### 6.4. Environmental Characteristics

Note that fine, cloudy, and rainy weather conditions were identified as the significant determinants affecting injury levels.

Fine weather was statistically significant in 2016–2017 G2 multi-vehicle freeway rear-end crashes, posing potential spatiotemporal instability and non-transferability by crash type. The cloudy weather tended to decrease the no injury likelihood and increase the minor injury likelihood in 2016–2017 G2 multi-vehicle (marginal effects of −0.0187 and 0.0210 in Table A6) and 2014–2015/2016–2017 G25 two-vehicle freeway rear-end crashes (marginal effects of −0.0356/−0.0355 and 0.0375/0.0338 in Table A7), whereas the marginal effects showed inconsistent influences on severe injury likelihood. In addition, rainy weather conditions were positively linked to minor injury likelihood and negatively related to no/severe injury likelihood in 2014–2015 multi-vehicle and 2018–2019 two-vehicle/multi-vehicle G25 freeway rear-end crashes (positive marginal effects of 0.0458/0.0232/0.0471 defined for minor injury in Table A7). These inconsistent findings might be attributed to the risk-compensation psychology of drivers being more cautious and conservative when encountering reduced visibility or wet surfaces [28,52]. Similarly, the interesting findings can be seen in the study of Yan et al. [29]. Examining the crash-injury severities in adverse weather, the detrimental driving conditions such as slippery pavement surface, illustrated complicated and opposite impacts on the crash outcomes over different periods. Several empirical studies have also confirmed that the inclement driving environments (slippery roadway surface, poor light conditions, and low visibility) stemming from adverse weather conditions (cloudiness, rain, snow, fog, and sleet) might trigger more severe outcomes. Nonetheless, other research efforts have shown that adverse weather reduces the likelihood of severe outcomes [53,54].

### 6.5. Temporal Characteristics

Significant temporal determinants were identified as daytime, weekday, and season time. The marginal effects in Table A6 and Table A7 suggested considerable variations by time, freeway, and crash type. More severe injury crashes tended to occur on Monday in 2018–2019 G2/G25 two-vehicle freeway rear-end crashes, whereas Monday increased the minor injury levels by 3.70% in 2016–2017 G25 multi-vehicle freeway rear-end crashes. Tuesday was observed to increase the severe injury likelihood of 2016–2017 G2 two-vehicle/multi-vehicle freeway rear-end crashes. Saturday was positively associated with severe injury outcomes in 2018–2019 G2 two-vehicle/multi-vehicle (marginal effects of 0.0234/0.0024 in Table A6) and 2014–2015 G25 multi-vehicle freeway rear-end crashes (marginal effect of 0.0056 in Table A7). This finding is as expected, as the free-flow conditions on weekends might lead to excessive speed [29]. Furthermore, frequent alcohol consumption on weekends could be an explanation [55].

As for time of day, early morning (24:00–05:59) and evening (18:00–23:59) were found to increase the likelihood of minor or severe injuries (see Table A6 and Table A7). This finding is as expected due to the lower visibility during early morning and evening, and drivers tend to suffer from speeding and fatigue/drowsy driving more frequently during these time periods [56]. The afternoon was associated with a lower likelihood of severe injury outcomes in 2016–2017 G2 two-vehicle freeway rear-end crashes, because the drivers’ vision is better during the afternoon [57].

In terms of season, spring was found to reduce the likelihood of severe injury in 2016-2017 G2 two-vehicle/multi-vehicle freeway rear-end crashes. However, summer tended to increase the probabilities of minor/severe injury outcomes in 2016–2017 G25 two-vehicle freeway rear-end crashes (marginal effects of 0.0192/0.0009 in Table A7). This finding is in line with previous research efforts [58], in which the authors suggested that adverse weather conditions (e.g., typhoons and rainstorms) that typically occur in summer can significantly deteriorate the driving environment. In 2014–2015 G25 multi-vehicle freeway rear-end crashes, autumn tended to increase the minor injury likelihood by 6.55%. In addition, winter was related to a higher risk of minor injury outcomes in 2014–2015 G2 multi-vehicle (marginal effect of 0.0129 in Table A6) and severe injury outcomes in 2014–2015 G25 two-vehicle freeway rear-end crashes (marginal effect of 0.0191 in Table A7).

### 6.6. Spatial Characteristics

The estimation results suggested that the significant indicators varied among the G2 and G25 models, suggesting possible spatial instability and non-transferability by crash type (The bridges in the G2 Freeway are mainly class bridges across channels with steeper grades). In addition, the potential reason for spatial instability in the interchange indicator might be the shorter average distances (5.25 km) between interchanges in G25 compared to that (13.59 km) in G2.

Bridge segments were observed to decrease the minor injury likelihood and increase the severe injury likelihood, specifying the higher crash risk of the segments in G2 two-vehicle freeway rear-end crashes in the years 2014–2015/2018–2019 (see Table A6). This finding can be explained by the shorter sight distance corresponding to steeper grades in bridge segments, which renders less time for the drivers to operate timely and properly [57,59].

Moreover, in G25 multi-vehicle freeway rear-end crashes in years 2014–2015/2018–2019, interchange segments tended to increase the minor injury likelihood and decrease the severe injury likelihood (marginal effects of 0.0548/0.0030 and −0.0087/−0.0062 in Table A7). This finding might also be explained by the risk-compensation psychology of drivers when negotiating the interchange segments, which decreases the severe risk propensity [28].

### 6.7. Traffic Characteristics

Traffic volumes were observed to be statistically significant in the majority of these models, while the marginal effects suggested possible instability in the influences on minor injury likelihood. Showing spatiotemporal stability and similarities in different crash types, the greater AADT was involved with the higher possibility of no injury and lower possibility of severe injury in G2 2014–2015/2018–2019 two-vehicle and 2014–2015/2016–2017 multi-vehicle (see Table A6) and G25 2016–2017/2018–2019 two-vehicle and 2014–2015/2018–2019 multi-vehicle rear-end crashes (see Table A7). As pointed out by previous research efforts, high traffic volumes might contribute to lower travel speeds [56,57,60].

### 6.8. Crash Characteristics

In the case of Emergency Medical Service (EMS), only the arrival time of (20–60 min) was significant in 2018–2019 G2 two-vehicle freeway rear-end crashes, whereas the three EMS indicators produced significance in G25 models. Overall, the marginal effects indicated that the response time had positive effects on the injury levels (see Table A6 and Table A7), consistent with recent evidence [58,59].

### 6.9. Random Parameters and Heterogeneity in Means and Variances

In G2 models, three variables were identified as random parameters under normal distributions, including Tuesday specific to no injury, spring specific to severe injury, and AADT specific to severe injury. In G25 models, rain was specific to no injury, evening was specific to minor injury, $Ls_{min}$ specific to no injury, and bridge specific to minor injury were statistically significant random parameters. Tuesday for instance, this variable was identified as a random parameter in 2014–2015 G2 two-vehicle freeway rear-end crashes. The mean (*standard deviation*) of 1.298 (*−1.128*) indicated that on Tuesday the likelihood of no injury likelihood increased for 12.5% of the observations and decreased for the other 87.5% of the observations.

Regarding heterogeneity in the mean of the random parameter, speeding was found to increase the mean of the Tuesday indicator in G2 models. In the presence of a speeding indicator, the Tuesday tended to decrease the likelihood of no injury by 1.4% in 2014–2015 G2 two-vehicle freeway rear-end crashes. The Spring in 2016–2017 increased the severe injury likelihood by 0.4% of G2 two-vehicle freeway rear-end crashes. For G25 models, the presence of speeding under rainy conditions decreased the no injury likelihood by 6.7% in 2018–2019 multi-vehicle freeway rear-end crashes. On Sunday, evening time increased the minor injury likelihood by 0.4% in 2014–2015 G25 two-vehicle freeway rear-end crashes. In the presence of rainy weather conditions, the $Ls_{min}$ decreased the no injury likelihood by 5.2% in 2016–2017 G25 two-vehicle freeway rear-end crashes, whereas the bridge segments increased the minor injury likelihood by 0.3% in 2014–2015 multi-vehicle freeway rear-end crashes.

With respect to heterogeneity in variances of the random parameter, during the early morning on Tuesday, the no injury likelihood decreased by 0.8% for 2014–2015 G2 two-vehicle freeway rear-end crashes. Likewise, under rainy weather conditions, spring increased the severe injury likelihood by 0.1% in 2016–2017 G2 two-vehicle freeway rear-end crashes. In G25 models, when it comes to passenger cars the rainy weather increased the no injury likelihood by 8.6% in 2018–2019 multi-vehicle freeway rear-end crashes. In the case of speeding behaviors, the evening tended to increase the minor injury likelihood by 0.2% in 2014–2015 G25 two-vehicle freeway rear-end crashes.

Overall, the interactions of these variables suggested a higher risk of speeding indicator in the early morning, evening time, and rainy weather conditions.

## 7. Conclusions

To examine transferability and heterogeneity for crash type (two-vehicle versus multi-vehicle freeway rear-end crashes) and spatiotemporal stability of determinants affecting the injury severity, this study modeled the data of crashes in Beijing-Shanghai Freeway (G2) and Changchun-Shenzhen Freeway (G25) over 2014–2019. Accommodating the heterogeneity in the means and variances, the random parameters logit model was proposed to estimate three potential crash injury severity outcomes (no injury, minor injury, and severe injury) and identify the determinants in terms of the driver, vehicle, roadway, environment, temporal, spatial, traffic, and crash characteristics. The likelihood ratio tests illustrated that the effects of factors varied significantly across crash type, time, and freeway. Significant variations were observed in the marginal effects of determinants between two-vehicle and multi-vehicle freeway rear-end crashes, and spatiotemporal instability was reported in several determinants, including truck and early morning. In addition, the heterogeneity in means and variances of random parameters, revealing the interactions of random parameters and other insignificant variables, suggested the higher risk of speeding indicator in the early morning, evening, and rainy weather conditions.

Certainly, the temporal and spatial instability have not been exactly explained in the recent research efforts. However, the importance of exploring the spatiotemporal instability of factors affecting the injury severity has been addressed in the current study, whereas biased and inaccurate estimated results and recommendations might suffer from ignoring the spatiotemporal instability. The current finding could help freeway designers, decision-makers, and management strategies to understand the contributing mechanisms of the factors to develop effective management strategies and measurements.

The current study still has some limitations. First, sociodemographic characteristics of drivers, actions of drivers, and other social attributes were not recorded in the datasets. Second, data from more freeways and for longer periods should be collected to explore more accurately estimated results and where the structural breaks are. Meanwhile, the general applicability of the model in this study would be verified. Third, more advanced statistical models should be developed to account for heterogeneity and yield more reliable results. Finally, the expansion projects will be reconstructed on the two freeways, which are supposed to be finished by 2023. It might be more evidential to explore the pre and after differences in the crash severities, which will be conducted in further research.

## Figures and Tables

**Figure 1 ijerph-19-10282-f001:**
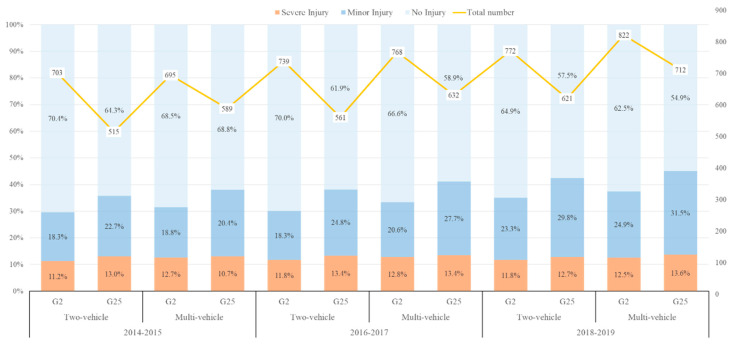
Two-vehicle and multi-vehicle freeway rear-end crash injury severity distribution for G2 and G25 freeways over the years: 2014–2019.

**Table 1 ijerph-19-10282-t001:** A summary of findings in previous research efforts regarding rear-end crashes.

Variable Names	Findings
** *Driver characteristics* **	
Gender	Inconsistent findings have been demonstrated about the effects of gender on the injury severity in different types of crashes [12,15,20]. However, a limited body of studies exclusively analyzed the effects of rear-end crashes on injury severity. For instance, Zhang and Hassan [8] indicated that male drivers increased the possibility of fatal rear-end crashes compared to female drivers.
Age	In the research efforts of Chen et al. [10] and Chen et al. [19], age showed statistically insignificance in rear-end crashes, whereas Zhang and Hassan [8] demonstrated that young drivers are related to fatal outcomes in rear-end crashes. Yu et al. [7] illustrated that young (<25) and middle-aged (25–60) drivers tend to cause less severe injuries in rear-end crashes in work zones.
Alcohol or medicine	The involvement of alcohol or medicine significantly increased the frequency of more severe injuries in rear-end crashes [7,11].
** *Vehicle characteristics* **	
Vehicle type	Heavy vehicles were found to be associated with more severe injury outcomes. For example, the involvement of trucks increased the possibility of more severe injury outcomes [10,11]. Heavy trucks were also found to be significant in predicting drivers’ fatalities [19]. Passenger cars increased the possibility of injury, whereas sports utility vehicles only increased the possibility of property damage [7].
Number of vehicles	Previous studies also indicated inconsistences on the effects of the number of vehicles. Two-vehicle crashes are identified as the most common rear-end type causing fatalities [19], while Yu et al. [7] reported that two-vehicle collisions lead to a lower possibility of severe-injury outcomes in the rear-end crashes.
** *Roadway characteristics* **	
Roadway geometry	More severe injury severity outcomes occurred on the curved segments [8]. The probability of rear-end crashes is related to the length of the longitudinal slope [21].
Speed limit	Speeding was statistically significant in fatal crashes in work zones, whereas the higher speed limit was related to severe outcomes in rear-end crashes [8]. Yu et al. [7] also reported that the speed limit is positively related to injury and possible injury possibility.
Number of lanes	Two-lane roadways were positively related to the fatalities in rear-end crashes [19].
** *Environmental characteristics* **	
Weather condition	As expected, rainfall conditions increase the severity levels of rear-end crashes [9], and windy weather is related to more severe injury outcomes for occupants in rear-end crashes [10,19]. However, foggy weather tends to mitigate the injury severity in work zone rear-end crashes [8].
Pavement condition	A counterintuitive finding was reported by Qi et al. [11], in which the authors demonstrated that rear-end collisions occurring on slippery roadways caused less severe outcomes.
** *Temporal characteristics* **	
Time of day	The propensity of daytime rear-end crashes is distinctly higher than that during the night [2,22]. The dusk and dawn time tends to decrease slightly the likelihood of injury [10].
Weekdays	Driving at night on weekends was strongly associated with injury and fatal outcomes in rear-end collisions [8].
** *Traffic characteristics* **	
Traffic volume	The average daily traffic volumes significantly affect the occurrences of urban rear-end crashes [23]. Weng et al. [24] also reported that greater traffic volume will increase the risk propensity of work zone rear-end crashes. Wang et al. [6] illustrated that the average annual daily traffic volumes are positively related to severe and fatal rear-end crashes.

**Table 2 ijerph-19-10282-t002:** A summary of methodological approaches used for analysis on rear-end crashes.

Methodological Approaches	Previous Research
** *Statistical methods* **	
Nested logit model	Abdel-Aty and Abdelwahab [33]
Stepwise regression	Meng and Weng [35]
Ordered probit model	Ghasemzadeh and Ahmed [36]
Random-parameters ordered probit model	Zhang and Hassan [8]
Mixed probit model	Weng et al. [24]
Markov switching multinomial logit model	Malyshkina and Mannering [37]
Random-parameters logit with heterogeneity in means and variances	Yu et al. [7]
** *Data-driven methods* **	
Binary classification tree and logistic regression models	Yan et al. [22]
Decision table/Naïve Bayes (DTNB) hybrid classifier	Chen et al. [19]
Support vector machine and mixed logit model	Ahmadi et al. [38]
Decision Tree Approach	Champahom et al. [39]

**Table 3 ijerph-19-10282-t003:** Two-vehicle and multi-vehicle rear-end crashes statistics in Beijing-Shanghai Freeway (G2) and Changchun-Shenzhen Freeway (G25) over 2014–2019.

Subgroup		Severe Injury		Minor Injury		No Injury		Total	
		G2	G25	G2	G25	G2	G25	G2	G25
2014–2015	Two-Vehicle	79	67	129	117	495	331	703	515
	Multi-Vehicle	88	77	131	147	476	365	695	589
2016–2017	Two-Vehicle	87	75	135	139	517	347	739	561
	Multi-Vehicle	98	85	158	175	512	372	768	632
2018–2019	Two-Vehicle	91	79	180	185	501	357	772	621
	Multi-Vehicle	103	97	205	224	514	391	822	712

**Table 4 ijerph-19-10282-t004:** Results of LRT across year periods (two-vehicle G2 rear-end crash models).

*y* _1_	*y* _2_		
	2014–2015	2016–2017	2018–2019
2014–2015	–	82.20 (15) [>99.99%]	112.39 (10) [>99.99%]
2016–2017	145.05 (12) [>99.99%]	–	91.74 (10) [>99.99%]
2018–2019	135.49 (12) [>99.99%]	206.01 (15) [>99.99%]	–

**Table 5 ijerph-19-10282-t005:** Results of LRT across year periods (multi-vehicle G2 rear-end crash models).

*y* _1_	*y* _2_		
	2014–2015	2016–2017	2018–2019
2014–2015	–	85.64 (14) [>99.99%]	66.37 (7) [>99.99%]
2016–2017	160.78 (9) [>99.99%]	–	168.66 (7) [>99.99%]
2018–2019	297.63 (9) [>99.99%]	426.85 (14) [>99.99%]	–

**Table 6 ijerph-19-10282-t006:** Results of LRT across year periods (two-vehicle G25 rear-end crash models).

*y* _1_	*y* _2_		
	2014–2015	2016–2017	2018–2019
2014–2015	–	102.90 (10) [>99.99%]	110.33 (12) [>99.99%]
2016–2017	108.56 (13) [>99.99%]	–	124.83 (12) [>99.99%]
2018–2019	144.84 (13) [>99.99%]	135.62 (10) [>99.99%]	–

**Table 7 ijerph-19-10282-t007:** Results of LRT between different year periods (multi-vehicle G25 rear-end crash models).

*y* _1_	*y* _2_		
	2014–2015	2016–2017	2018–2019
2014–2015	–	121.28 (9) [>99.99%]	104.96 (16) [>99.99%]
2016–2017	74.94 (10) [>99.99%]	–	79.71 (16) [>99.99%]
2018–2019	147.83 (10) [>99.99%]	119.47 (9) [>99.99%]	–

## Data Availability

The data used to support the findings of this study have not been made available because the crash dataset is obtained through the traffic police department and the administrative department. The data cannot be disclosed due to confidentiality requirements.

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
