# Peer review of "Assessment of Two-Vehicle and Multi-Vehicle Freeway Rear-End Crashes in China: Accommodating Spatiotemporal Shifts"

_ijerph, 2022, doi:10.3390/ijerph191610282_

Round 1
Reviewer 1 Report
The paper presents a statistical analysis of the effects of many variables (i.e., driver, vehicle, roadway, environment, temporal, spatial, traffic, crash characteristics) on two-vehicle versus multi-vehicle freeway rear-end crashes. The random parameters logit model was employed to estimate three potential crash injury severity outcomes (no, minor or severe injury). The data collected regarded two China freeways in the period from 2014 to 2019.
The quality of the presentation is poor and must be improved. Since there is a large amount of data involving many variables, a selection should be conducted and the results should be presented in a more simplified, focused and intuitive manner. Then, an Appendix section with all the data can be added for further details.
-The Introduction section should be more focused on the rear-end crashes. Maybe the Introduction section could be delete and few sentences can be added in the Literature Review section.
-As suggested by the Journal, the references should be indicated in the manuscript with the paragraph number.
-All the Tables from number 4 in the manuscript present too much data. Either than in the manuscript, consider to add an Appendix. In the manuscript, only the relevant data should be selected and reported. Moreover, a more intuitive and simple way to present the data is the use of e.g. histograms.
-In table 4 and 5, the st.dev in parenthesis is usually higher than the related value, and negative values are reached. How is it possible? Please explain. In the manuscript, it is not explained how these values have been calculated.
-All the results in Lines 239-269 are not presented in a proper manner. As abovestated, there are too much data, and the reader don't know where to focus on.
-As for Table 4 and 5, also Tables 10-11-12-13-14 contain too much data, that could be reported in an Appendix. Whereas the relevant should be selected and reported in the manuscript.
-In e.g. Table 12- time of day should be between 1 and 0, whereas the value is negative. Many cases, with the range between 1 and 0, reported similar values. Check the data, or better explain how they have been calculated.
-In the discussion paragraphs, which is the most important, i.e. 6.1, 6.2,...., 6.9, the reference to the corresponding Table should be indicated. The affirmations must be based on the statistical number reported in the tables, but the reader don't understand where to watch on.
Reviewer 2 Report
1.Throughout the text, there are a number of formulations that may have ambiguous meaning, this requires corrections.
2. The wording of the concept of stability is used in many places. The stability of technical systems is evaluated according to the Lyapunov criterion. If the concept has a different meaning then it needs to be defined.
3. Throughout the article, the concept of frequency of events and probability is not distinguished. I remind you that probability is the integral of the probability density function, that is, you need to know the probability density function to determine probability.
4. On page 9 different density functions are given, it would be necessary to state what method these functions were determined from experimental data and at what level of significance.
5. In line 199, the concept of maximum likelihood is given. It is necessary to specify how it was determined.
6. In line 175 the concept of stochastic is used - this is completely false. A stochastic process is defined on two sets, in the set of elementary events and in the set of definiteness. This definition does not correspond to the formulation used.
7.In Formula 1, Sij is not a probability density function, therefore, probability cannot be defined based on it.
8.I suggest that you use terms that have their reference in mathematical relations. All the aforementioned elements need to be corrected.
9.From the experience of many countries, the problem of rear-end collisions on highways is also the result of so-called "bumper-to-bumper" driving. Some countries have made it mandatory to maintain distances between vehicles (inspections are carried out, mainly with the use of drones, and non-compliance is punishable by heavy fines). This is a comment on assumptions, which were not considered in the work.
Reviewer 3 Report
The paper proposes random parameters logit models with heterogeneity in means and variances to examine the contributing factors of crash using data from Beijing‐Shanghai Freeway and Changchun‐Shenzhen Freeway during 2014 to 2019. The Authors have presented a comprehensive work supported with very extensive data.
To enhance the paper the following comments and suggestions may be considered for revising the paper.
1. The Abstract presents an extensive discussion about the topic, yet seems to be unclear distinguishing problem statement, proposed solution, method, results and contribution(s) of the research. Please re-arrange the abstract so that it clearly describes the above sections. The codes of G2 and G25 are better not to be in abstract, since they are described later in the section of “Data Description”.
2. The paper presents comprehensive literature review, discussion regarding the topic from different perspectives, including methodological review, supported by many references. However, the data and review take a dominant portion of the paper. More portion on result and discussion is preferred.
3. To make the paper more convenience for reader, it is suggested that only important data and results are presented in the main part of the paper. Some descriptions and interpretations accompanying the data and results are necessary. Some supporting data may be given in appendices.
4. A number of factors influencing the severity of rear crash for different conditions are described. These are presented numerically in Tables followed by some interpretations. However, the effect of every factor on the severity of injury may not be easily grasped. Describing the relations more clearly in the form of descriptions explaining the data will make the discussion more interesting. Repeating and narrating the data in the discussion is discouraged.
5. The expression “As shown in Table 3 and Figure 1 reported three injury severity classifications in the dataset” (line 150,151) seems to be non-standard expression. Please consider revision.
6. The Authors may consider replacing the word “outcome” for the term of Injury since it tends to have the positive meaning. The word like “effect” may be more suitable.
Round 2
Reviewer 1 Report
The authors changed many aspects of the manuscript, highlighting the results or better explaining some findings. The manuscript can be accepted in the present form,
Author Response
Thank very much for the comments.
Reviewer 2 Report
The authors in response to comments in many cases either did not understand the comment or gave the wrong answer.
There was a request that the concept of stability be defined so that it is not associated with Lapunov stability. Using the term stability can be understood as Lapunov stability. It needs to be unambiguously defined that it has nothing to do with this stability.
There was a suggestion to explicitly define the concept of event frequency and probability. Recall that the probability of an event xi is
where:
P(xi) - probability of occurrence of event xi.
P(x) - probability density function - analytical form determined on the basis of real events, using χ2 or λ or other test.
Only quantities determined in this way can be called probabilities.
It is reported that Sij determines the probability of injury. Please state how this probability was determined. Otherwise, there is no way to check the correctness.
It is common to use the term significance level α=0.05. As opposed to the confidence interval, which for such a significance level will be 0.95.
In answer number 7, the authors consider Sij to be the probability density. Please provide an analytical relationship for the probability.
(Using the literature without providing analytical formulas is a mistake).
The answer in formula (5) gives an integral that has no limits. Drink what independent variable does it have? From the analytical form on the right, you should think that the independent variable is β. The magnitude of β should vary from a minimum to a maximum. Without this element, it is not clear how to determine such a function. In addition, in formula (5) f(β/) is a density function - please provide the analytical relationship for this function.
It would also be good to specify why the logarithmic form was adopted for the reliability function. The reference to item [17] is not very convincing.
